# Cryo-EM structure of a bacteriophage M13 mini variant

Qi Jia[1] & Ye Xiang [1] ✉

Filamentous bacteriophages package their circular, single stranded DNA genome with the major coat protein pVIII and the minor coat proteins pIII, pVII, pVI, and pIX. Here, we report the cryo-EM structure of a ~500 Å long bacteriophage M13 mini variant. The distal ends of the mini phage are sealed by two cap-like complexes composed of the minor coat proteins. The top cap complex consists of pVII and pIX, both exhibiting a single helix structure. Arg33 of pVII and Glu29 of pIX, located on the inner surface of the cap, play a key role in recognizing the genome packaging signal. The bottom cap complex is formed by the hook-like structures of pIII and pVI, arranged in helix barrels. Most of the inner ssDNA genome adopts a double helix structure with a similar pitch to that of the A-form double-stranded DNA. These findings provide insights into the assembly of filamentous bacteriophages.

Filamentous bacteriophages are typically between 1–2 µm in length and 6–7 nm in diameter. The capsids of filamentous bacteriophages encapsidate a circular single-stranded DNA genome. The length of the virion is determined by the size of the encapsidated genome[1,2]. The micrometer long capsids of filamentous bacteriophages are assembled from thousands copies of the major coat protein pVIII and a few copies of the four minor coat proteins, including pIII, pVI, pVII, and pIX[3,4].

Unlike other bacteriophages, filamentous phages are released from their host cells without causing lysis, but through a process similar to the secretion of virulence factors[5–7]. During this process, a complex of pVII and pIX on the cytoplasmic side of the inner membrane recognizes the packaging signal on the genome and initiates the genome packaging[8]. Both pIX and pVII are small proteins and have 32 and 33 residues, respectively. Driven by a phage encoded ATPase[9], the phage particles extrude from the inner membrane with the major coat protein pVIII enwrapping the genome with its C-terminal segment[10]. The precursor of the major protein pVIII contains 73 amino acids, with the first 23 N-terminal residues functioning as a signal peptide that directs the insertion of the precursor pVIII into the membrane[11]. During genome packaging, the signal peptide on the precursor pVIII is cleaved by a signal peptidase, releasing the mature pVIII from the membrane[12]. The mature pVIII, which contains the C-terminal 50 residues of the precursor pVIII, assembles to form the capsid. The genome packaging is terminated by adding a complex of pIII and pVI to the distal end of

the particle when a packaging termination signal is sensed. The minor coat protein pIII is produced with an 18-amino acid signal peptide, which is cleaved to form a mature pIII of 406 residues after being inserted into the membrane[13]. The minor coat protein pVI has 112 amino acids and no post-translational processing has been reported for pVI. Although structures of different filamentous phages have been extensively studied[14–17], the structures of the complexes formed by the minor coat proteins and mechanisms involved in the initiation and termination of filamentous phage genome packaging are poorly understood.

The filamentous bacteriophage M13 is commonly used for phage display, in which the target gene of interest is ligated into the phage gene *III*[18]. In addition to its function in terminating genome packaging, the protein encoded by gene *III* (pIII) also functions as the receptor binding protein of filamentous bacteriophages[19,20]. Here, we report the structural studies of a mini variant of the bacteriophage M13. Through these studies, we were able to build an atomic model of the entire mini filamentous M13 phage and reveal the structure of the single-stranded DNA genome that is encapsidated by the mini M13 phage.

## Results

### Production and cryo-EM reconstruction of the M13 mini phage
To investigate the mechanisms involved in the assembly of filamentous phages, we constructed a mini variant of M13 by using a

[1]Beijing Frontier Research Center for Biological Structure, Center for Infectious Disease Research, SXMU-Tsinghua Collaborative Innovation Center for Frontier Medicine, Department of Basic Medical Sciences, School of Medicine, Tsinghua University, Beijing 100084, P.R. China.
✉e-mail: yxiang@mail.tsinghua.edu.cn

modified genome that contains a functional replication origin, a packaging signal sequence and a dysfunctional replication origin[21] (Supplementary Fig. 1A). The genome of the miniphage consists of only 221 nucleotides[21] (Supplementary Fig. 1A). With the assistance of a helper phage that provides all the required phage proteins for the life cycle, the mini genome could be packaged, resulting in the production of a mini phage of ~50 nm in length (Supplementary Fig. 1A–D). We purified the mini M13 phage (Supplementary Fig. 1B, C), and a cryo-EM reconstruction of the purified mini M13 phage was calculated, yielding an EM map with an overall resolution of 3.5 Å with 5-fold symmetry imposed (Fig. 1A; Supplementary Fig. 2 and Supplementary Fig. 3; Supplementary Table 1). However, the local resolutions at the distal ends of the particle are poor (Supplementary Fig. 3B). The results show that the mini phage is 495 Å long, and the capsid consists of 25 protein layers, with each layer containing five molecules. Protein layers 1–5 of the capsid assemble to form a spherical cap (Fig. 1). Protein layers 6–23 of the capsid form a cylinder with a diameter of ~60 Å. The molecules in the cylinder are arranged in helical arrays. The twist and rise of the helical array are ~36° and ~16 Å, respectively, which are similar to the reported parameters for the helical array of the major coat protein pVIII in previous studies[15]. Protein layers 24 and 25 of the capsid form a 150 Å long sharp cap at the distal end (Fig. 1).

We further performed local refinements of the mini phage by dividing the particle into the top, middle and bottom three segments. Reconstructions of the three segments were calculated separately with 5-fold symmetry imposed, resulting in maps of the three segments at resolutions of 3.5, 3.1, and 3.3 Å, respectively (Supplementary Fig. 3B, C and Supplementary Fig. 4; Supplementary Table 1). By combining the reconstructions of the segments and the entire particle, we built a complete atomic model of the particle, which consists of 105 copies of the major coat protein pVIII and all the minor coat proteins, including 5 pVII molecules, 5 pIX molecules, 5 pVI molecules and 5 pIII molecules. The pVII and pIX molecules constitute the first and the second layer of the particle, respectively (Fig. 1B). The 105 major coat protein pVIII molecules constitute the protein layers 3–23 of the particle. The pIII and pVI molecules constitute the protein layers 24 and 25 of the particle, respectively. Densities for the N-terminal residues 1–261 of pIII, the N-terminal residues 1–4 of pVII and pVIII, residues 1 and 111–112 of pVI are missing and the corresponding models were not built. The pVIII, pVII, and pIX molecules have similar structures that contain a long helix. The pVIII molecules in the top and bottom showed different conformations compared to those in the central segment (Fig. 1A). Based on the copy number of the major coat protein pVIII and the size of the genome, the calculated ratio of the nucleotides to the pVIII capsid units is 2.1.

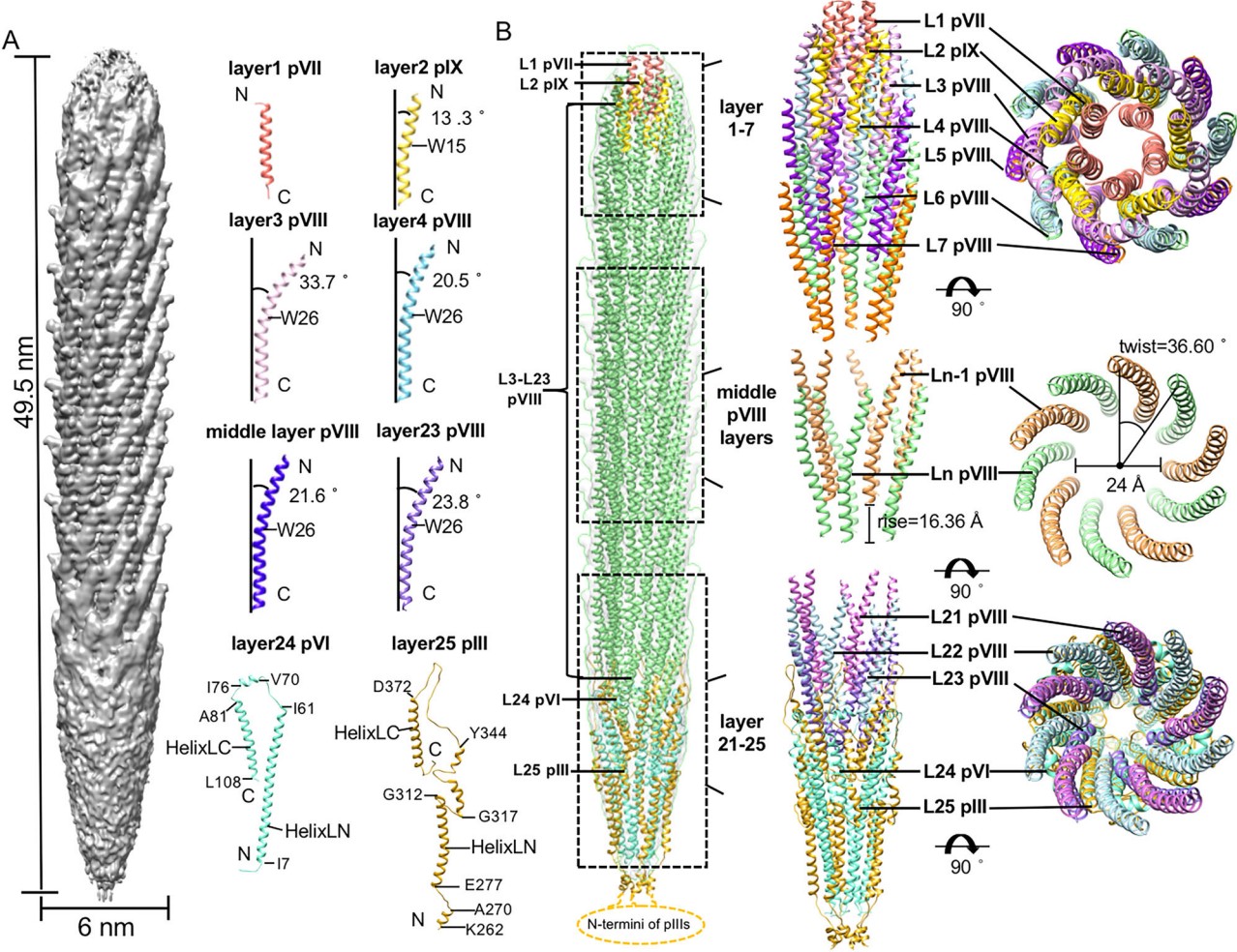

**Fig. 1 | The mini phage structure. A** Left: A surface-rendered representation of the cryo-EM density map showing the structure of the entire mini phage. Right: ribbon diagrams of the coat proteins showing the monomers from different protein layers. The contouring level of the density map was set at 0.00392 e⁻/Å³. **B** Left: Ribbon diagrams showing the structure of the entire mini phage. The major coat protein pVIII is colored light green, the four of minor proteins are colored in salmon (pVII), gold (pIX), goldenrod (pIII), and aquamarine (pVI), respectively. Right: Ribbon diagrams showing the top, middle, bottom segment structures. The coat proteins from the same protein layer are colored the same.

## Structure of the top cap

The top spheric cap is a complex of pIX, pVII, and pVIII (Fig. 1B). The helices of pIX and pVII have a similar length of ~45 Å. The five pVII molecules form a symmetric plug-like pentamer that are located at the top middle of the cap. The pVII helices are straight and almost parallel to the long axis of the particles with a small tilt angle of 2.6° to the main axis of the particle (Fig. 1A). The small tilt angle of the helices results in a clockwise arrangement of the five pVII molecules when viewed from the top (Fig. 1B). The five pVIIs mainly interact with each other through hydrophobic residues, including F7, I10, A13, M14, L21, A24, L25, and A29. At the bottom of the pVII helices, five hydrogen bonds are formed (pVII I28$^{O}$-pVII' Q32$^{NE2}$, distance: 2.9 Å) to further stabilize the helix bundle (Fig. 2A). The side chains of five F7 residues point towards the particle's main axis, forming a hydrophobic ring at the top. The outer surface of the pVII pentamer is distinctly amphiphilic. The bottom half of the outer surface is surrounded by the five pIX molecules of layer 2 and is rich in hydrophobic residues (Fig. 2A, B), while the exposed top half is hydrophilic and has several negatively charged residues, including D6 and D8 (Fig. 2A). The 5 pIX molecules of the protein layer 2 assemble around the pVII pentamer (Fig. 1B). Each pIX is partially embedded in the cleft between two pVIIs. Interactions between pIX and pVII mainly involve hydrophobic interactions, hydrogen bonds, and salt bridges (Fig. 2B, C, Supplementary Table 2). Both pVII and pIX have a hydrophobic segment of 19 residues in pIX and 25 residues in pVII, with lengths of approximately 32 Å and 39 Å, respectively (Fig. 2D).

The C-terminal region of the pIX helix, comprising approximately one-third of the helix, is straight and nearly parallel to the main axis of the virion. The helix of pIX bends around residue W15, introducing a curvature similar to those of the pVIII helices[15] (Fig. 1A). The bend creates an angle of ~13° between the N-terminal portion of the pIX helix and the main axis of the virion (Fig. 1A). The molecules from the first layer of pVIII (layer 3) are inserted in between the pIX molecules and have a completely different contact environment compared to the pVIII molecules from other layers. The long helix of pVIII bends in the middle around residue W26. The bent angle (~34° from the main axis of the virion) of the pVIII in layer 3 is significantly larger than those (~21°) of the pVIIIs in layers 4−5 and the middle part of the particle, due to the specific environment of pVIII molecules in the layer for establishing interactions with pIX and pVII (Fig. 1A and Fig. 2C). In addition, orientations of the side chains involved in interactions are quite different in the pVIIIs of layer 3 when compared to those from other layers (Fig. 2E). The bent helices of pIX and pVIII have an anticlockwise arrangement around the pVII pentamer (Fig. 1B).

The cap constituted by pVII, pIX, and pVIII has a unique lumen that is decorated with four positively charged residues, including R33 of pVII, K40, K43, K48 of pVIII and one negatively charged residue, E29 of pIX (Fig. 3A). Among these, residues R33 of pVII and E29 of pIX are in proximity to the encapsidated DNA genome (Fig. 3B), indicating that these residues could be involved in direct interactions and specific recognition of the genome packaging signal, which was supposed to have a hairpin structure[8]. The side chains of other positively charged residues, including K40, K43, and K48 of pVIII, point towards the DNA density, indicating that these residues may also interact with the phosphate-sugar backbone of the hairpin packaging signal structure. In addition, the densities around the distal ends of R33 of pVII and E29 of pIX are weak in the reconstruction, which may be due to the asymmetric interactions of these residues with the genomic DNA inside. Further mutagenesis studies of residues E29 of pIX and R33 of pVII showed that these residues are essential for the assembly of living viruses. The mutant pVII-R33A causes an order of magnitude decrease in phage titer, while the mutant pIX-E29A does not release any progeny phage from the host cells (Supplementary Fig. 5).

## Structure of the middle pVIII array

The middle part of the M13 miniphage consists of 20 protein layers (from layer 4 to layer 23) of pVIII assembling to form a cylindrical capsid with an outer diameter of ~60 Å and an inner diameter of ~24 Å. Similar to the pVIII assembly in the wild-type phage, the pVIII molecules in the layers overlap with their neighboring subunits, resembling fish scales, and form right-handed spirals in the symmetric cylindrical shell (Fig. 1B). The helical parameters of the pVIII array determined by the real space search with Relion[22], are 36.60° for the twist and 16.36 Å for the helical rise (Fig. 1B), which are similar to the values of 36.40° for the twist and 16.60 Å for the rise determined by solid-state NMR[15]. Further helical averaging of the reconstruction improved the overall resolution of the EM map to 2.7 Å (Supplementary Fig. 3B, C). Each pVIII interacts with eight neighboring pVIII proteins in six different layers primarily through hydrophobic interactions, hydrogen bonds and salt bridges (Supplementary Fig. 6, Supplementary Table 2).

## Structure of the bottom cap

Our reconstruction of the bottom cap reveals a pencil-tip-like structure (Fig. 1B), where five pIIIs and five pVIs form a sharp cap enwrapping the pVIII cylinder. Both pIII and pVI have a hook-like structure with two long helices (helixLN and helixLC), located at the N- and C-termini of the structures, respectively (Figs. 1A and 4A, B). In addition to these two long helices, pIII has a short helix at the N-terminus, and two short helices (middle short helices) and a long loop situated in between the two long helices (Figs. 1A and Fig. 4A), while pVI has only a short helix in between the two long helices (Figs. 1A and 4B). The C-terminus of pIII is buried in the bottom segment structure and is involved in accommodating the C-terminal ends of the pVIIIs in the final pVIII layer (Figs. 1B and 5). Residues I400 and R402 at the C-terminus of pIIIs form intermolecular hydrogen bonds (pIII I400$^{O}$-pIII' R402$^{NH1}$, distance: 2.5 Å) that tether the distal ends of the C-terminal helices. Both helixLCs and helixLNs of pIIIs are well separated (Fig. 4A). The helixLC of pIII interacts mainly with the helixLC of pVI and the two middle short helices of pIII (Figs. 1B and 5A). The contacts with the two middle short helices primarily occur through hydrophobic residues and one hydrogen bond (pIII L344$^{O}$-pIII' Y386$^{OH}$, distance: 2.6 Å), which is situated in the middle of helixLC from one pIII and links the two middle short helices from a neighboring pIII. The helixLNs of pIII interact mainly with the helixLNs of pVI (Fig. 5A). The N-terminal short helices of pIIIs assemble into a helix barrel, maintained by hydrophobic interactions and hydrogen bonds (Fig. 4A, Supplementary Table 2). PVI interacts with adjacent pVIs through hydrophobic residues and three hydrogen bonds that bridge the N-terminal long helix of one pVI and the C-terminal long helix of a neighboring pVI (Fig. 4B, Supplementary Table 2).

The helixLNs of pIIIs and pVIs tighten at the bottom of the cap and alternately distribute with intervals, forming a sharp-tip like helix barrel (Fig. 1B). In this structure, the helixLNs from pIIIs are located at the outer positions while the helixLNs of pVIs occupy the inner positions. PIII establishes 11 hydrogen bonds and six salt bridges with two neighboring pVIs (Fig. 5A, Supplementary Table 2). In the sharp-tip like helix barrel, the outer surface of pVI is predominantly hydrophobic but contains two positively charged rings that are formed by R12, K30, and K31 in the helixLN. However, the helixLN of pIII is rich in negatively charged residues, which are in close proximity to the positively charged regions of pVI and shield the hydrophobic surface of pVI. The arrangement of pIII and pVI leads to the outer surface of the helix barrel being rich in charged residues (Fig. 5B).

The C-terminal long helices together with the loops or short helices that link the N- and C- terminal long helices of pIIIs and pVIs enwrap around the distal end of the pVIII array and facilitate the interactions with the molecules from the last three layers of pVIII (Fig. 6A). Interactions between pIII, pVI, and pVIII primarily involve hydrophobic residues (Fig. 6B). The long loop 354−372 of pIII extends

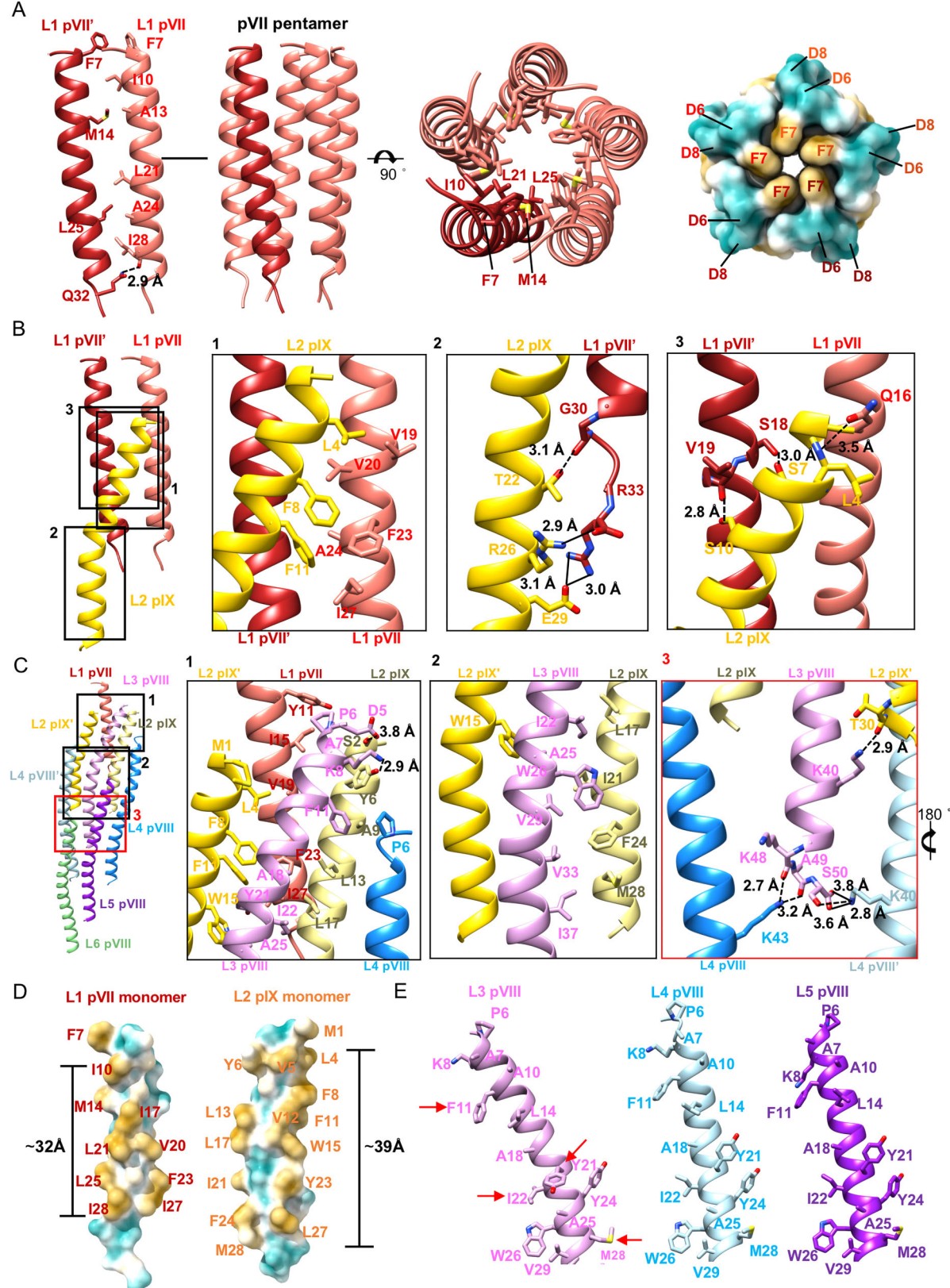

upward for 25 Å and interacts with the last three layers of pVIIIs. The C-terminal distal ends of pVIIIs in the last pVIII layer exhibit a shift towards the particle's main axis compared to those of pVIIIs from other layers (Fig. 6A). The middle part of the pIII long loop forms a β-hairpin structure, rich in aromatic residues, which inserts into a pocket formed by residues of three pVIII helices from layers 21, 22,

and 23. In addition to the hydrophobic interactions, residues K363 and Y365 at the top of the β hairpin forms a salt bridge and a hydrogen bond, respectively, with E20 of pVIII from layer 23 (pIII K363$^{NZ}$-pVIII E20$^{OE1}$, distance: 3.5 Å, pIII Y365$^{OH}$-pVIII E20$^{OE1}$, distance: 3.5 Å) (Fig. 6B; Supplementary Table 3). Except for a slight shift in the side chains of Y21 and Y24, the pocket that accommodates the loop

**Fig. 2 | Structures and detailed interactions in the top cap. A** Ribbon diagrams and surface-rendered representations showing the symmetric plug-like pVII pentamer in side and top views. In the ribbon diagrams, one pVII molecule is colored wine red, while other four pVII moleucles are colored salmon pink. Residues involved in the key contacts are shown in sticks with the C, N, O atoms colored salmon/wine red, blue and red, respectively. The surface is colored cyan for hydrophilic residues and mustard yellow for hydrophobic residues. **B** Ribbon diagrams showing the interactions between on pIX and two pVIIs. The two pVII molecules are colored salmon pink and wine red, respectively. Residues involved in the key contacts are shown in sticks with the N, O atoms colored blue and red, respectively. The C atoms in the sticks are colored the same as the corresponding ribbon diagrams. The hydrogen bonds and salt bridges are represented with dash lines and solid lines, respectively. **C** Ribbon diagrams showing the detailed interactions between the pVIIIs in layers 3–6 and the pVII-pIX complex. The zoom-in views at the right showing detailed contacts at different interfaces. The key contacting residues are shown in sticks with the N, O atoms colored salmon, blue and red, respectively. The C atoms in the sticks are colored the same as the corresponding ribbon diagrams. **D** Surface-rendered representations showing the hydrophobic segments of pVII and pIX. Hydrophilic residues are colored cyan and hydrophobic residues are colored mustard yellow. **E** Ribbon diagrams showing the key residues involved in the hydrophobic interactions among pVIIIs in layers 3–5. PVIII in layer 3 is colored plum, pVIII in layer 4 is colored light blue and pVIII in layer 5 is colored purple. The key contacting residues are shown in sticks with the N, O atoms colored blue and red, respectively. The C atoms in the sticks are colored the same as the corresponding ribbon diagrams.

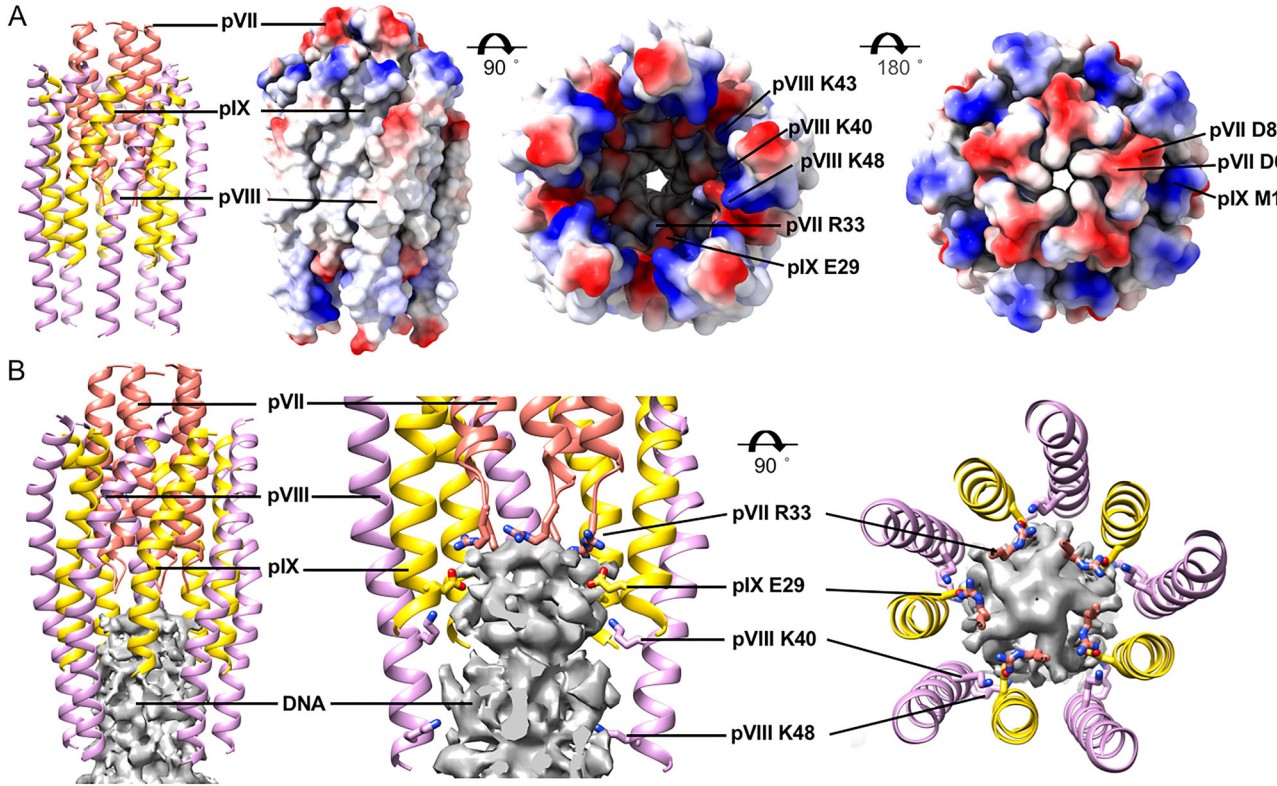

**Fig. 3 | Structure of the top cap and its interaction with the inner genome. A** Ribbon diagrams and surface-rendered representations showing the structure of the top cap. PVIIs are colored salmon, pIXs are colored gold and pVIIIs in layer 3 are colored plum. The surface in side and top views is colored according to the surface electrostatic potential, with red for negative potential and blue for positive potential. **B** Ribbon diagrams of the coat proteins and surface-rendered representations of the density map of the inner genome DNA showing the contacts of the residues in the lumen with the inner genome DNA. Residues involved in direct contacts with the genome DNA are shown in sticks with the N, O atoms colored blue and red, respectively. The C atoms in the sticks are colored the same as the corresponding ribbon diagrams.

of pIII exhibits no significant difference compared to those in the middle segment, where the pockets are occupied by pVIII (Fig. 6C). However, in comparison to the interactions with pVIII at the same location, the interactions with pIII are more extensive, indicating that pIII may be able to effectively compete with pVIII for binding the pocket (Fig. 6B, C, Supplementary Table 3). The N-terminal region, including N1 and N2 domains of pIII, which are used as the ligation site for phage display and plays a role in the infection process, are completely disordered in the map[19,23,24].

### Structure of the circular ssDNA genome
To determine the structure of the inner ssDNA genome, we performed further classifications using the middle segment by searching for the orientation among the five equivalent positions[14,25,26]. The structure of the inner DNA genome can only be determined when it has a fixed relationship with the capsid. After several round of classifications, we observed a right-handed double helix structure of the DNA genome in the middle segment. The two strands are well separated and presumably correspond to the phosphate backbones of the DNA genome (Fig. 7A, B). However, densities for the bases are missing in the reconstructions. The double helix structure in the middle segment has four complete turns. The pitch of the double helix structure is ~28 Å for the top two turns and is ~31 Å for the bottom two turns. The diameter of the double helix structure is ~19–24 Å, which depends on the contouring level set for the map (Fig. 7C). The pitch is consistent with that of the A-form dsDNA, which has a pitch of ~28 Å and a diameter of ~23 Å. The diameter of the A-form dsDNA measured with the centers of the backbones is ~13 Å, which is consistent with the diameter measured from the centers of the strands in the double helix structure we obtained (Fig. 7B). Our observation of the right-handed double helix

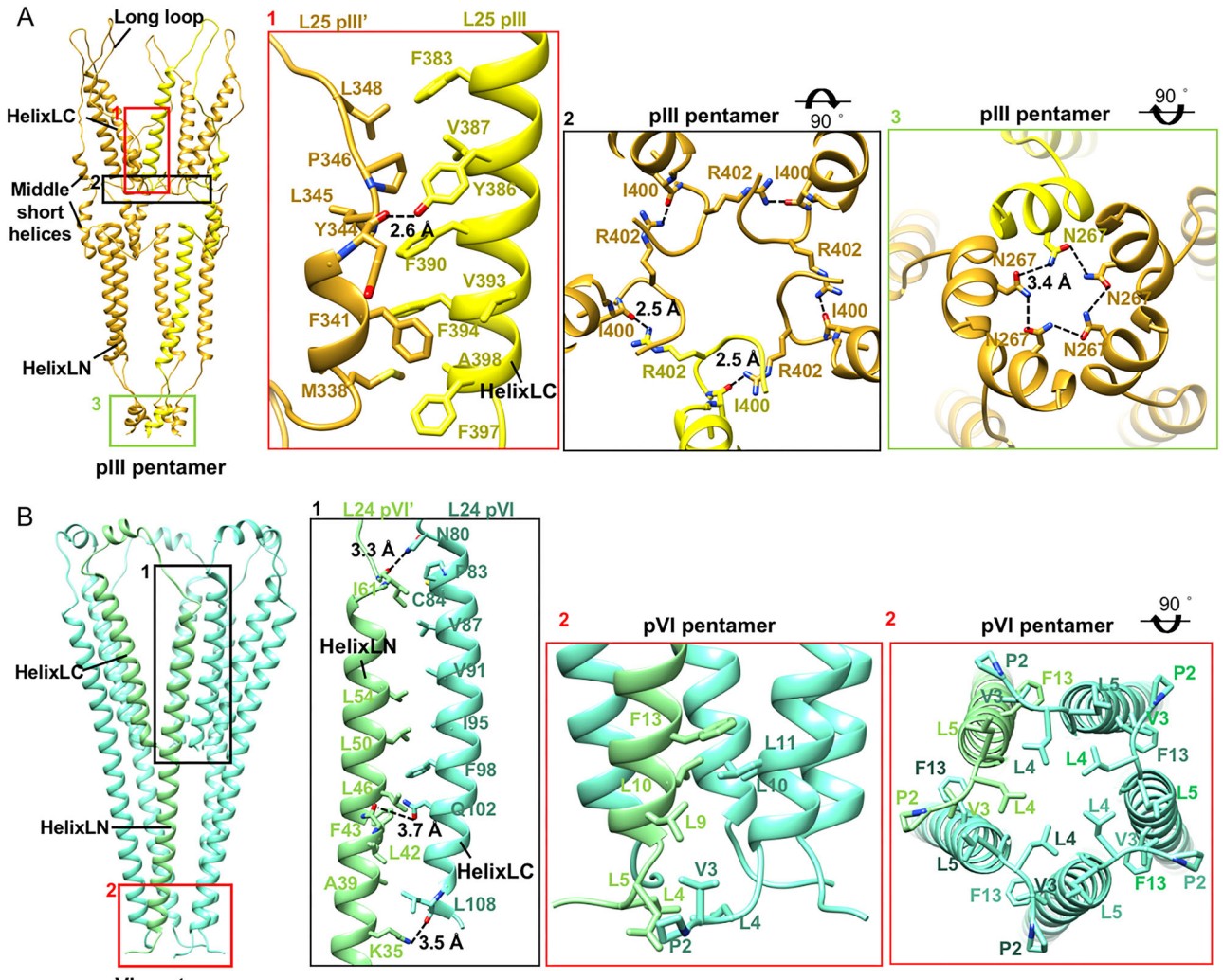

**Fig. 4 | Structures and arrangements of pVI and pIII in the bottom cap. A** Ribbon diagrams showing layer 25 of the pIII pentamer. One pIII molecule is in light yellow, while other four pIII molecules are in goldenrod. The zoom-in views at the right showing detailed contacts at different locations. Residues involved in the key contacts are shown in sticks with the N, O atoms colored blue and red. The C atoms in the sticks are colored the same as the corresponding ribbon diagrams. The hydrogen bonds are represented with dash lines. **B** Ribbon diagrams showing layer 24 of the pVI pentamer. One pVI is in light green, while other four pVI molecules are in aquamarine. The zoom-in views at the right showing detailed contacts at different contacting interfaces. Residues involved in the key contacts are shown in sticks with the N, O atoms colored blue and red. The C atoms in the sticks are colored the same as the corresponding ribbon diagrams. The hydrogen bonds are represented with dash lines.

structure of the genome is consist with previous studies on Ff[27]. We then extended the box of the particles and calculated the asymmetric structure of the entire particle with the orientation determined with the central fragment (Fig. 7A and Supplementary Fig. 2). The results showed that densities corresponding to the inner genome DNA is ~350 Å long. The segment with clear double helix structure starts from layer 8 of pVIII, ends around layer 16 of pVIII, and has a length of ~140 Å. The DNA segments at the top and bottom of the particle do not have the double helix feature and have a length of ~100 Å and 110 Å, respectively (Fig. 7A).

Assuming that all the nucleotides of the genome adopt a double helix conformation with similar parameters as these of the A-form dsDNA, which has a pitch of ~28 Å and 11 bp per turn, a total of ~269 nts could be encapsidated in the capsid. The middle segment, which has the A-form dsDNA-like double helix structure, could have packaged 110 nts. However, the lumen of the capsid is not big enough to encapsidate the 221 nts of the genome if it is in any other conformation, such as a double helix structure that has similar parameters as those of the norm B-form dsDNA conformation (Supplementary

Table 4). Although clear features of a double helix could not be observation for the bottom DNA segment, the densities exhibit distinct repeating helical features with a pitch of ~38 Å. This suggests that the bottom part of the genome may possess a stretching double helix structure when compared to those of a A/B-form dsDNA (Fig. 7B). Assuming that the ssDNA genome in the bottom segment adopts a B-form dsDNA-like double helix structure, it could potentially accommodate a total of 65 nts. Although having not been observed in any of the filamentous bacteriophages, the stretching B-form dsDNA-like structure was reported in many other protein-DNA complexes, such as in the bacterial RecA-DNA complex[28].

The top segment of the genome adopts a conformation similar to an unwound double helix, where the two strands of the helix are almost parallel to each other (Fig. 7B). This conformation is reminiscent of observations made with a B-form dsDNA pulled using atomic force microscopy, resulting in a flat conformation with two parallel strands[29]. Based on the calculations, the top segment is estimated to package ~46 nts and has a length of ~100 Å, yielding a rise of 4.5 Å per base pair. In the pulling assay, a 12 bp B-form dsDNA would have a

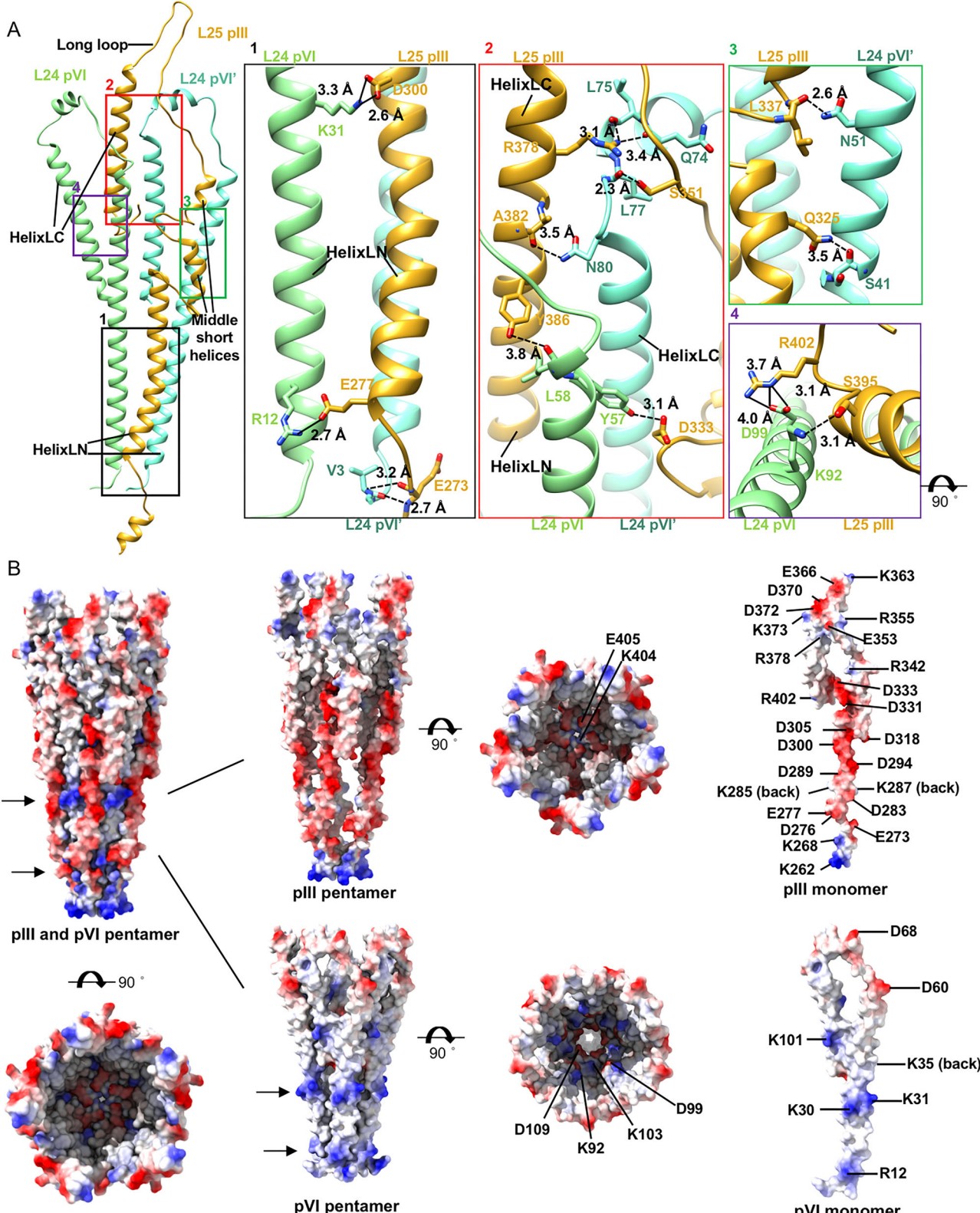

**Fig. 5 | Interactions between pVI and pIII in the bottom tip. A** Left: Ribbon diagrams showing the interactions between one pIII and two pVIs. PIIIs are colored goldenrod and the two pVIs are colored aquamarine and light green, respectively. Right: zoom-in views at right showing the hydrogen bonds and salt bridges. Residues involved in the key contacts are shown in sticks with the N, O atoms colored blue and red. The C atoms in the sticks are colored the same as the corresponding ribbon diagrams. The hydrogen bonds and salt bridges are represented with dash lines and solid lines, respectively. **B** Surface-rendered representations of the bottom cap and the protein layers and individual structural components of the bottom cap. The surface is colored according to the surface electrostatic potential, with red for negative potential and blue for positive potential. The arrows indicate the two positively charged rings of pVIs.

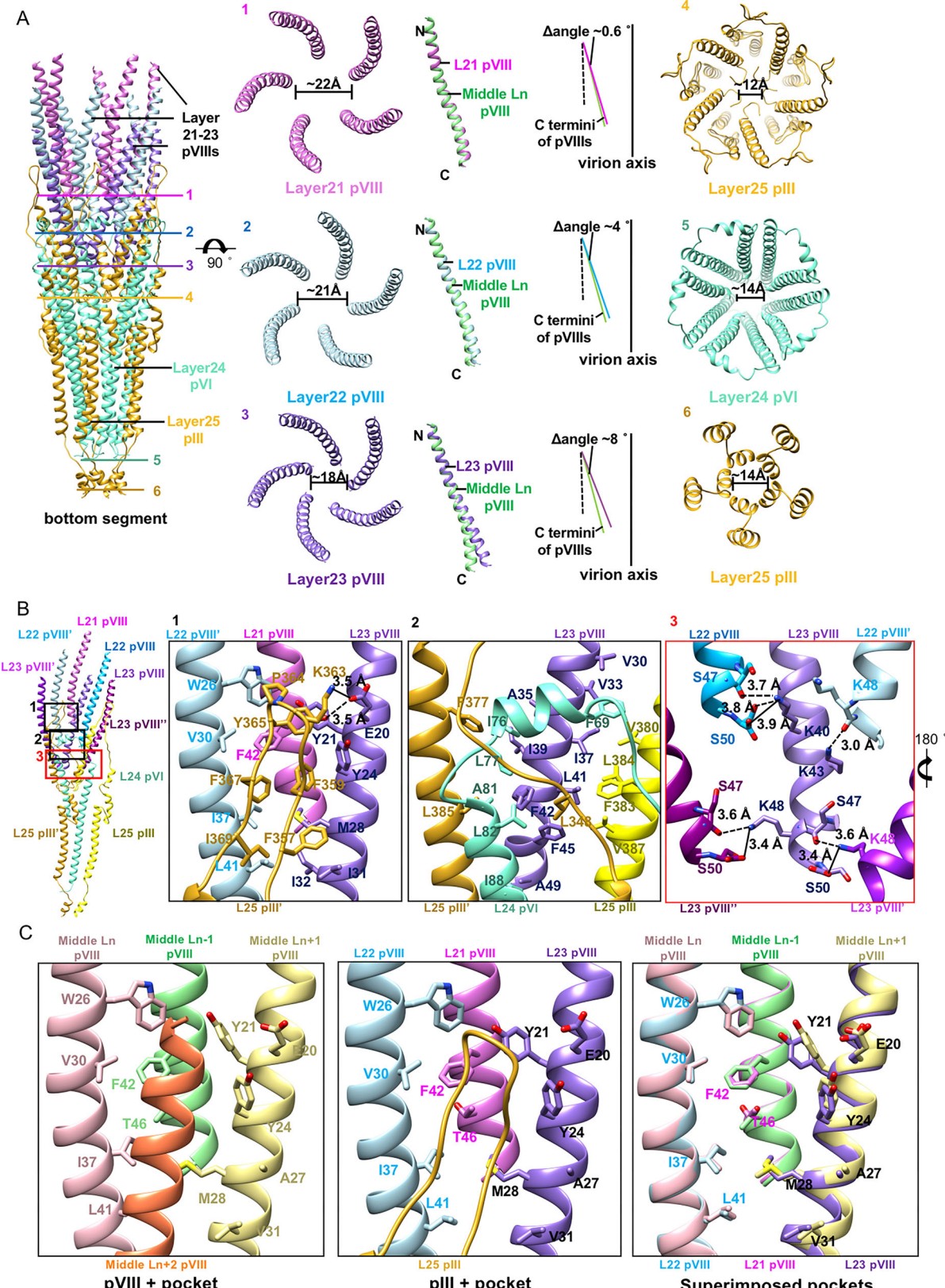

length of 84 Å when it is completely unwound, corresponding to a rise of 7.6 Å per base pair. Consequently, the top segment of the genome may adopt a conformation that is similar to the partially unwound dsDNA. Previous studies suggested that the packaging signal may have a hairpin structure that serves as a marker for the DNA encapsidation[30,31].

## Interactions between the capsid and the ssDNA genome

The inner surface of the capsid is decorated with the positively charged residues K40, K43, K44, and K48, which are clustered in one region of pVIII (Fig. 8A). The distribution of the positively charge region in the capsid follows the symmetry of the capsid (Fig. 8A), which, however, is not consistent with that of either the A- or B-form

**Fig. 6 | Structure and arrangements of the layers in the bottom cap. A** Left: Ribbon diagrams of the bottom cap showing the structures and arrangements of different layers. Layers 21–23 of pVIII are colored orchid, light blue and middle purple, respectively. Layer 24 of pVI and layer 25 of pIII are colored in goldenrod and aquamarine, respectively. Right: structural comparisons of the channels in different layers and the pVIIIs in layer 21–23 with pVIII in middle layer. Layers 21–23 of pVIII and the middle layer pVIII are colored orchid, light blue, middle purple and light green, respectively. The C terminus of pVIII in layer 23 has a significant shift towards the virion axis. **B** Ribbon diagrams showing the detailed interactions

between the pVIIIs in layers 21–23 and the pIII-pVI complex. The zoom-in views at the right showing detailed contacts at different interfaces. The key contacting residues are shown in sticks with the N, O atoms colored blue and red, respectively. The C atoms in the sticks are colored the same as the corresponding ribbon diagrams. **C** Left and Middle: Ribbon diagrams and structural comparisons of the pockets with pVIII (left) and with pIII (middle). Right: Ribbon diagrams showing the comparisons of the key residues in the pocket that accommodates pIII with these in the pocket that accommodates pVIII.

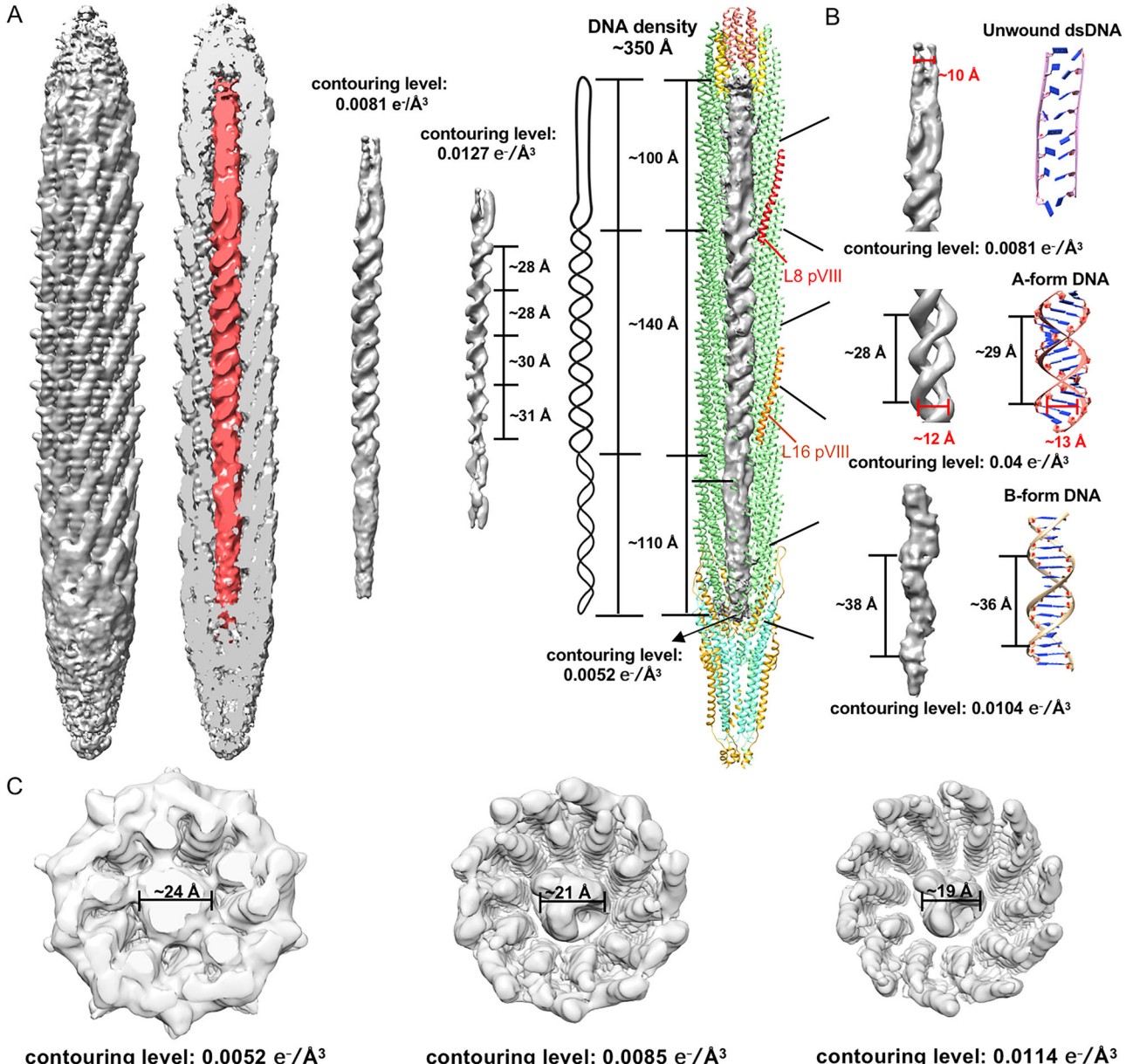

**Fig. 7 | Structure of the inner circular ssDNA genome. A** Left: surface-rendered representations of the density map of the mini phage and calculated without imposing any symmetry showing the structure and organization of the inner circular ssDNA genome. The contouring level of the density map of the mini phage was set at 0.0044 e⁻/Å³. Middle: surface-rendered representations of the inner circular ssDNA genome at different contouring levels. A schematic diagram of the DNA structure model is shown on the right. Right: Ribbon diagrams of the coat proteins and surface-rendered representations of the density maps of the inner ssDNA genome showing the structure and organization of the inner circular ssDNA genome. **B** Left: Surface-rendered representations showing the top, middle, and bottom segments of the inner circular ssDNA genome. Right: Schematic diagrams showing the structures of an unwound dsDNA fragment, an A-form dsDNA fragment and an ideal B-form dsDNA fragment. **C** Surface-rendered representations of the density maps of the middle segment at different contouring levels.

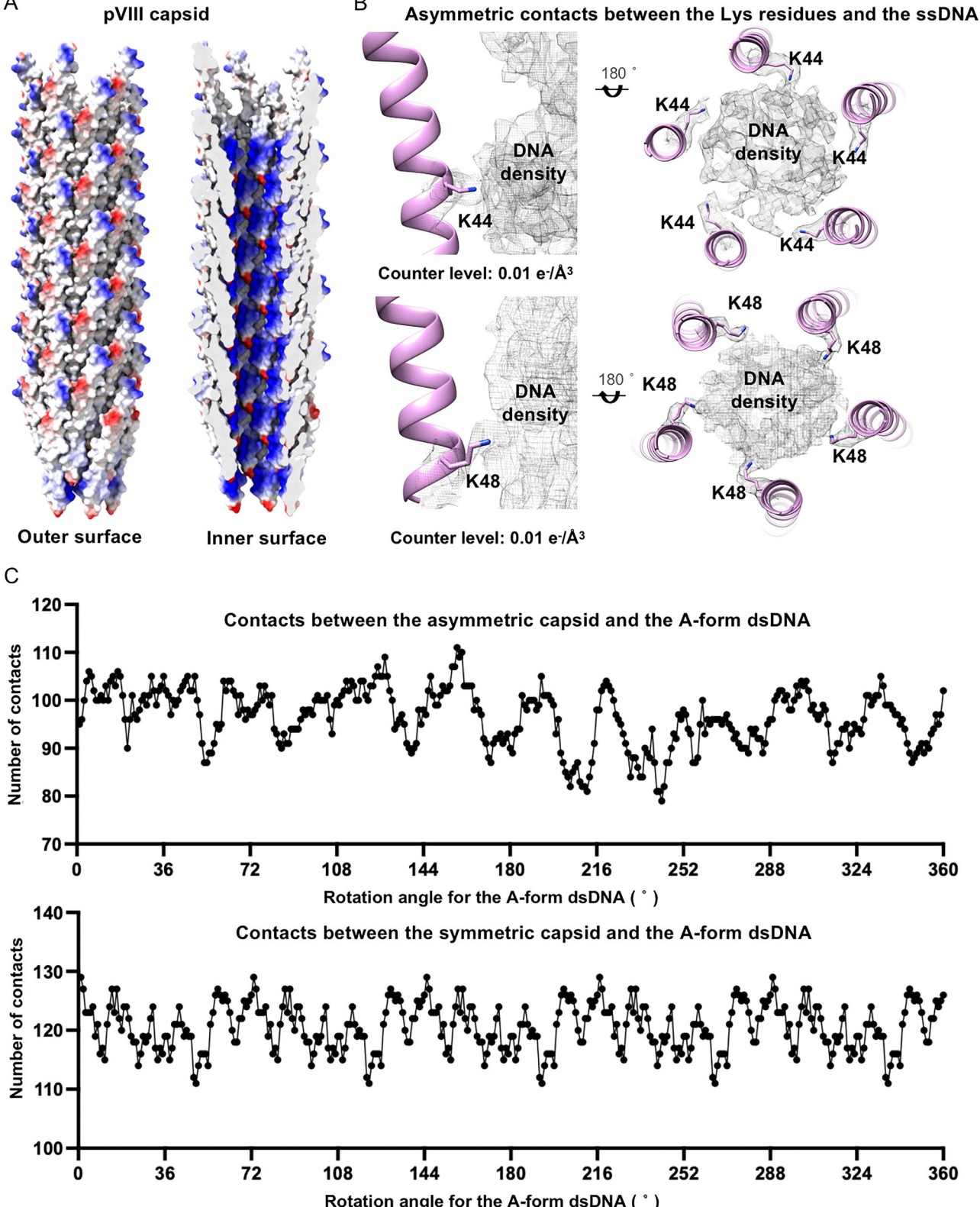

**Fig. 8 | Interactions between the inner circular ssDNA genome and the capsid.** **A** Surface-rendered representations of the density maps of the capsid. The surface is colored according to the surface electrostatic potential, with red for negative potential and blue for positive potential. Left: the outer surface of the capsid; right: a cross section of the capsid showing the inner surface. **B** Ribbon diagrams of the capsid in the asymmetric density map showing the asymmetric interactions between the inner surface Lys residues and the ssDNA genome. **C** Plots showing the number of contacts at each rotation position of an A-form dsDNA in the capsid. Top: the number of contacts between the asymmetric capsid and the A-form dsDNA; bottom: the number of contacts between the symmetric capsid and the A-form dsDNA. Only the contacts between the NZ atoms of the inner surface Lys residues and O3, OP1, and OP2 atoms of the phosphate groups are considered. The cutoff distance used is 4 Å.

dsDNA. Analysis of the asymmetric density map of the middle segment showed that K43, K44, and K48 of some pVIIIs in layers 8–16 have significant asymmetric features and are in close proximity to the densities of the ssDNA (Fig. 8B), suggesting possible direct interactions of these Lys residues with presumably the phosphate groups of the ssDNA phosphate-sugar backbone. The asymmetric conformation also indicates possible induced conformational changes of the inner surface Lys residues upon binding the ssDNA. To figure out possible interactions between the ssDNA genome and the capsid, we placed an A-form dsDNA structure of 4 turns in the capsid of the middle segment that contains layers 8–16 of pVIII. The length of the DNA (114.4 Å) is approximately the same as that of the capsid segment (114.8 Å). Then, the dsDNA model was rotated around the long axis with a step size of 1 degree. Close contacts (< or = 4 Å) between the NZ atoms of the inner surface Lys residues and the OP1, OP2, and O3 atoms of the phosphate groups were counted at each rotation position (Fig. 8C). The results showed that the average number of contacts at a rotation position is ~100. At some positions, the ssDNA has a larger number of contacts with the capsid. However, the number of contacts varies when a symmetric capsid structure built with the helical averaged map was used (Fig. 8C). Furthermore, the plot calculated with the symmetric capsid shows no distinct peaks, indicating that the capsid with a uniform and symmetric conformation is unlikely able to establish a unique interaction with the ssDNA and trap the ssDNA at a specific position.

Similarly, a B-form-like dsDNA placed in the capsid can also establish extensive interactions with the capsid. Analysis of the 5-fold symmetry-averaged density maps of the bottom and top segments, where the DNA adopts significantly different conformations compared to that in the middle segment, revealed that the Lys residues adopt similar conformations as those in the middle segment (Supplementary Fig. 7A). In the asymmetric reconstruction of the bottom segment, some of the Lys residues also display asymmetric features. These data combined suggest that the capsid is flexible for adapting genome DNA in various different conformations through likely induced conformational changes of the inner surface Lys residues. In our reconstruction, the precise positions of the phosphate groups of the ssDNA can not be determined due to the limited map resolution. The relative position of the ssDNA genome to the capsid could be further investigated through other techniques, such as solid-state NMR.

## Discussion

In M13 phage display, proteins and peptides are predominantly through the minor coat protein pIII and the phage particles are enriched by specific binding to a target of interest[32–34]. The conventional approach involves adding the target protein on the N-terminal of pIII. Although the insertion of external genes has been tested on other coat proteins, these methods have largely relied on trial and error[35–38]. By leveraging the detailed high-resolution structure, it becomes possible to accurately identify insertion sites on the coat proteins. This advancement has the potential to enhance and broaden existing display systems, facilitating their development and expansion.

It has been proposed that pVII and pIX are responsible for anchoring to the membrane to facilitate specific recognition of the packaging signal and initiate the genome packaging[8,39]. Our models of pVII and pIX reveal that neither pVII nor pIX has a signal peptide for membrane anchoring. Both pVII and pIX contain a hydrophobic segment ~35 Å-long, which is sufficient for spanning the lipid membrane. However, a mechanism by which pVII and pIX are released from the membrane remains unclear. Our results demonstrate the essential role of charged residues in pVII and pIX for the assembly of functional virions, and the interactions between these charged residues and the packaging signal appear to be asymmetric. Moreover, extensive interactions exist in between pVII, pIX, and pVIII. These collective observations suggest that pVII, pIX, and pVIII may form a pre-assembled cap-like complex within the channel of pI and pXI, primed for specific recognition of the signal and initiation of genome packaging[8,40]. Alternatively, instead of directly anchoring on the membrane, pVII and pIX could indirectly anchor on the membrane through pVIII, which is known to anchor to the membrane through the secretase YidC and be released through the cleavage by a signal peptidase[11,41,42].

Unlike tailed bacteriophages, genome packaging of M13-like filamentous bacteriophages does not involve a preformed empty capsid. Instead, the ssDNA genome of M13 forms a complex with the non-structural protein pV in the cytoplasm. The packaging signal is exposed and not covered by pV, making it ready to be sensed by the top cap complex. However, the packaging signal in the top segment should have a different conformation from that observed in the absence of the top cap complex, as the stretching conformation of the ssDNA genome is extremely unstable without this complex. Similarly, the A-form dsDNA-like or the stretching B-form dsDNA-like double-helix structure could not be stable without the capsid. In the process of genome packaging, the major coat protein pVIII replaces pV and interacts with the ssDNA in a channel embedded in the membrane. The conformation of the genome could be determined by complex factors, including the preformed conformation in the genome-pV complex, the sequence of the genome, and the interactions with the coat proteins.

The extensive interactions between pIIIs and pVIIIs facilitate binding of pIIIs to the pockets formed by the last three layers of pVIII. However, the competitive binding of the pIII β hairpins to pVIIIs may be weak during normal packaging procedure due to the presence of unpacked genome DNA-pV complex exposed at the distal end. The exposed genome DNA-pV complex may disrupt the assembly of the helix barrel formed by the helixLCs of pIII and pVI, thereby destabilizing any bound pIII on the premature particles. Thus, termination of the genome packaging should occur after the last piece of the genome DNA is enwrapped by pVIIIs, creating favorable binding sites for pIIIs.

A-form dsDNA has been observed in a number of filamentous archaeal viruses that have a dsDNA genome[43–45]. The double helix structure of the ssDNA genome in the M13 miniphage exhibits important differences to the A-form dsDNA. In A-form dsDNA, the bases on the parallel strands are complementary to one another, whereas these in the M13 ssDNA are not due to the genomic sequence. This discrepancy may explain why we were unable to clearly observe densities corresponding to the bases in our reconstruction, as only the phosphate-sugar backbone was visible. The ssDNA should be stabilized by the capsid. The asymmetric interactions establish a unique relationship between the capsid and the ssDNA genome and stabilized the conformation of the ssDNA, which may have been preformed in the genome-pV complex.

The conformation of the genome in the mini phage also suggests that the genome may adopt different conformation at different locations of the wild-type virion. In facts, the genome in the wild-type phage could be even more compressed, as the reported nucleotide/pVIII ratio of wild-type phage falls within the range of 2.19–2.41[27], which is higher than the calculated ratio of 2.1 for the mini phage. The conformation of the genome in the wild-type phage may also include other conformations, as some studies have suggested that the inner genome may adopt a conformation similar to that of the base-inverted P-form dsDNA[46]. The P-form dsDNA-like conformation of the ssDNA genome must be stabilized by specific interactions with the capsid. The inner surface of the bacteriophage Pf1 contains positively charged residues that extend towards the center of the particles and interact with phosphate groups[47] (Supplementary Fig. 7B). The capsid of Pf1 does not have a 5-fold symmetry. Instead, the major coat proteins of Pf1 follow the helical symmetry of the P-form DNA genome. However, it is unlikely that the ssDNA genome in the bottom segment of the

M13 miniphage could adopt a structure resembling the P form dsDNA, as the positively charged residues in the lumen of the capsid maintain similar conformations to those in the middle segment (Supplementary Fig. 7A). In addition, the major coat proteins of M13 follow a helical symmetry that is completely different from that of the P-form DNA. During the peer review of this Article, a structure of the capsid assembly of the filamentous phage f1 was reported[48]. The capsid proteins of M13 and f1 are highly homologous and the symmetric structures of the M13 and f1 capsid assemblies have no substantial differences. However, structure of the genome DNA within the capsid of phage f1 was not determined by Conners et al.[48], which may be due to the symmetry imposed in the reconstruction or the heterogenous conformation of the genome DNA.

## Methods

### Mini phage production and sample preparation

Our system to produce mini phage is similar to a phage display system. The genome of the mini phage[21] was synthesized by the Qingnan company (Wuxi, China). The genome was cloned into the vector pUC19. The helper phage plasmid M13KO7 that encodes all the proteins required for the phage life cycle was generously provided by Dr. Lei Yin at Wuhan University. A 6×His tag was added after the 18-aa signal peptide of pIII by PCR. The helper phage plasmid M13KO7 with the 6×His tag-pIII was amplified by transformation into the host cell *E. coli* TG1, which was cultured in 2×YT (16 g tryptone, 10 g yeast extract, 5 g NaCl in 1 L) with 30 μg/ml kanamycin at 37 °C overnight. The helper phage produced was purified by PEG precipitation with 4% PEG8000 and 0.5 M NaCl.

To produce the mini phage, the plasmid pUC19-minigenome was transformed into *E. coli* TG1. When the cell grows to an OD600 = 0.5 in 2×YT with 100 μg/ml ampicillin, the helper phage M13KO7 with the 6×His tag-pIII was added at a M.O.I. of 0.01. The infected cells were cultured at 37 °C for 1 h and then 30 μg/ml kanamycin was added to the culture. The mini phages were produced by further culture of the infected cells overnight at 30 °C. The supernatant was collected by centrifugation and then applied to cobalt resin (Takara, #635653) for affinity purification (Supplementary Fig. 1B). The elution was analysis by SDS-PAGE gels and western blot with an anti-his tag antibody (Antibodys: CWBIO #CW0286M and #CW0102S used at the dilutions of 1:5000 and 1:10,000, respectively) (Supplementary Fig. 1B, C). In theory, the coat proteins translated by the helper phage genome can package both the genome of the mini phage and the genome of the helper phage, so the elution contains the long helper phage and the short mini phage. The elution from the cobalt resins was collected, concentrated and further purified by using a 5–20% sucrose density gradient, which was centrifugated for 18 h at 60,000 × *g* (JA sw41 Ti rotor, Beckman). Fractions 4 and 5 collected from the gradient contain the M13 mini phage, which were concentrated and checked by negative staining microscopy (500 μl/fraction). The concentrated sample was then used for cryo-EM grids preparation. Aliquots of 3.5 μl of mini phage at a concentration of 0.6–0.8 mg/ml were applied to glow discharged 200 mesh Quantifoil grids (1.2 μm hole size) or Quantifoil grids with 2 nm continuous carbon on top of the supporting film (1.2 μm hole size) using a Vitrobot Mark III (Thermo Fisher). The grids were blotted for 6.5 s in 100% humidity at 8 °C and were then immediately plunged into liquid ethane.

### Cryo-EM data collection

CryoEM data of the mini phage were collected at a nominal magnification of 29,000 (an effective pixel size of 0.97 Å) on an FEI Titan Krios electron microscope operating at 300 kV and equipped with a K2 Summit camera (Supplementary Fig. 1D; Supplementary Table 1). The defocus values were set in the range of −1.2 μm to −1.7 μm, and the total dose was ~40 electrons per Å². A total of 8127 micrographs were selected for further processing (Supplementary Fig. 2).

### Image processing

A total of 2,184,852 intact particles were picked with a box size of 640 pixels and the particles were subjected to 2D classifications (Supplementary Fig. 2). After 2D classifications, we picked 1,157,522 particles and re-extracted the selected particles with a box size of 240 pixels for the reconstruction of the middle segment. After several rounds of 3D classifications with C5 symmetry imposed, 368,282 particles were selected for the 3D auto-refinements, which yielded a density map at a resolution of 3.1 Å after post-processing, as indicated by the FSC curve (Supplementary Fig. 3C).

To determine the structure of the inner DNA core, asymmetric reconstructions of mini phage were performed by assuming a fixed relationship between the capsid and the inner genome. We re-extract the particles based on the orientations and centers obtained from high-resolution reconstruction of the middle segment, with a box size of 640 pixels (Supplementary Fig. 2). Then these particles were subjected to 2D classifications. The straight particles were selected and subjected to 3D classifications without orientation sampling. We chose one of classes and used the selected particles for 3D local auto-refine with C5 symmetry imposed. The reconstruction yielded a map with a resolution of 3.5 Å after post processing. We then re-extracted the same particles with a box size of 240 pixels. Orientations of the particles were expanded from C5 to C1. Five equivalent orientations of each particle were generated and the particles were subjected to 3D classifications without orientation sampling and symmetry imposed. A featureless cylindrical volume was used as an initial model for the 3D classifications. The 3D classification results showed clear helical features of the inner genome in one class. Local 3D auto-refinements with the selected class and C1 symmetry yielded a density map with a resolution of 3.4 Å (Supplementary Fig. 3B, C). Then, same orientations were applied to the intact particles, of which the reconstruction yielded a density map with a resolution of 3.6 Å (Supplementary Fig. 2 and Supplementary Fig. 3B, C).

To obtain the high-resolution structure of the distal ends, we used block-based reconstruction[49] and orientations of the middle segment to calculate the two distal ends of mini phage (Supplementary Fig. 2). The block-based reconstructions with C5 symmetry imposed resulted in a map at a resolution of 3.5 Å for the top segment (Supplementary Fig. 2 and Supplementary Fig. 3B, C), and a map at a resolution of 3.3 Å for the bottom segment (Supplementary Fig. 2 and Supplementary Fig. 3B, C). Further asymmetric reconstructions of the top and bottom segments resulted in a map at a resolution of 3.9 Å (Supplementary Fig. 2 and Supplementary Fig. 3B, C) for the bottom segment (Supplementary Fig. 2). We failed to have a high-resolution asymmetric reconstruction for the top segment. However, the inner genome at the top segment showed clear features in the asymmetric reconstruction of the entire particle (Fig. 7B).

### Construction of M13KO7 mutants

M13KO7 mutants were generated by using M13KO7 as a template with PCR. The construct pIX E29A was PCR-amplified from the M13KO7 plasmid by using the following primers: 5′-CCCGTTTAATGG-CAACTTCCTCATGAAAAAG-3′ (forwards) and 5′-CATGAGGAAGTTGC-CATTAAACGGGTAAAATAC-3′ (reverse). The construct pVII R33A was PCR-amplified from the M13KO7 plamsid by using the following primers: 5′-TGGGGGTCAAGCATGAGTGTTTTAGTG-3′ (forwards) and 5′-AAACACTCATGCTTGACCCCCAGC-3′ (reverse). After DpnI digestion, the PCR products were transformed into *E. coli* DH5α for plasmid amplifications. Then the plasmids with correct sequences were transformed into *E.coli* TG1 for producing mutant phages.

### Plaque assay of the M13KO7 mutants

Wild-type and mutant plasmids were transformed into *E.coli* TG1 cells for the production of phages. The supernatants collected were then used for infecting *E. coli* S2060 cells, which were generously

provided by Dr. Shuyi Zhang. *E. coli* S2060 contains a F plasmid with a phage shock promoter, which controls the expression of LacZ. Any phage infection will induce the expression of LacZ, which can be detected by X-gal and turn the infected cells into blue. The supernatant was properly diluted with medium and then 10 μl of the dilution was added to 90 μl S2060 cell culture, which was growing in the log phase and has an OD600 value of ~0.6. The infected cells were mixed with 1 ml LB that contains 0.7% soft agar. Then the mixture was plated on LB agar plates that contains 0.02% X-gal. The plates were incubated at 37 °C overnight and then the phage titers were determined by counting the blue plaques on the plates (Supplementary Fig. 5).

### Reporting summary

Further information on research design is available in the Nature Portfolio Reporting Summary linked to this article.

## Data availability

The atomic coordinates and EM maps have been deposited into the Protein Data Bank (http://www.pdb.org) and the EM Data Bank (http://www.emdataresource.org), respectively, with the accession numbers EMD-35795 (symmetric reconstruction of top segment), EMD-35793 (symmetric reconstruction of the middle segment), EMD-35794 (symmetric reconstruction of the bottom segment), EMD-35796 (asymmetric reconstruction of the middle segment), EMD-35797 (asymmetric reconstruction of the bottom segment), EMD-35798 (symmetric reconstruction of the full-length M13 mini variant), EMD-35805 (asymmetric reconstruction of the full-length M13 mini variant), 8IXL (atomic models of the symmetric top segment), 8IXJ) (atomic models of the symmetric middle segment), 8JWT (atomic models of the asymmetric middle segment), 8IXK (atomic models of the symmetric bottom segment). Source data are provided with this paper.

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

## Acknowledgements
We thank the Dr. Liang Huang for help with cryo-EM data collection and sample preparation. We thank the Tsinghua University Branch of the China National Center for Protein Sciences (Beijing) for providing the facility support. We thank Dr. Shuyi Zhang and Dr. Lei Yin for providing the materials, and Dr. Xinzheng Zhang for helpful discussions on the asymmetric reconstructions. Y.X. is funded by the Ministry of Science and Technology of China (grants: 2021YFA1300204), the Tsinghua University Vanke Special Fund for Public Health and Health Discipline Development (NO. 2022Z82WKJ013), the National Natural Science Foundation of China (grants: 31925023, 21827810, 31861143027), the Spring Breeze Fund of Tsinghua University, the Beijing Frontier Research Center for Biological Structure, and the Beijing Advanced Innovation Center for Structural Biology.

## Author contributions
Q.J. planned and performed the biochemical experiments, analyzed data, prepared the figures, and wrote the initial manuscript together with Y.X. Y.X. planned and supervised the experiments, analyzed the data, and wrote the manuscript.

## Competing interests
The authors declare no competing interests.
