## [Peer Review File · Nature Communications]

CryoEM structure of a bacteriophage M13 mini variantREVIEWER COMMENTS

Reviewer #1 (Remarks to the Author):

This is an impressive paper that provides new insights into the structure of filamentous bacteriophage which package ssDNA. A number of the observations are quite novel, such as the structure of the DNA in the phage. There are two major problems that need to be addressed in a major revision. One is that the paper needs to be carefully proofread by someone who speaks English as a native language. The other is that the discussion of the DNA structure could be expanded.

I began by indicating corrections needed, but gave up after the first page of the Introduction:

Line 38) “by thousands copies” should be “from thousands of copies”

Line 49) “function” should be “functioning”

Line 49) “which leads the insertion” should be “which directs the insertion”

Line 58) “have been extensively studies” should be “have been extensively studied”

On Line 61, “due to the extreme shape of the virion” makes no sense. I assume the authors are referring to the fact that the minor proteins are only found at the ends and thus would be lost in conventional helical structure determination.

Line 184, “could improve the overall resolution” should be “improved the overall resolution”, and the two have very different meanings.

The parameters for A- and B-form DNA are treated with great precision, such as “A-form dsDNA, which has a rise of 28.6 Å per turn” and in Fig. 7 B-DNA is labeled as 35.7 Å. In fact, there is an enormous literature on the considerable variability of these parameters for both A- and B-form. Further, they really should say the pitch here, and not the rise, as the rise is best used for describing the base pairs. Surprisingly, there is no discussion of how dsDNA has been observed packed in the A-form in a number of filamentous archaeal viruses, such as in Wang et al., PNAS 117, 19643 (2020). In fact, to the best of my knowledge, there are no structures of B-form dsDNA in any filamentous virus. While the authors cite micro-mechanical stretching of B-DNA in vitro, there are also many observations of proteins stretching B-DNA, such as the bacterial RecA protein (for example, Yang et al., Nature 586, 801, 2020).

Reviewer #2 (Remarks to the Author):

This paper describes the cryoEM structure of a mini-variant of the filamentous phage M13. The structure includes all the minor proteins of M13 and of the major coat protein pVIII. The ingenious trick to use an M13 mini-virus with a short filament allowed solving the complete structure of the virus. The structure uncovers the molecular details how the filament 5-fold symmetric pVIII lattice is capped by different proteins at its two extremities. The work is beautiful and an important step forward to understand M13 structure and function. This is of significant fundamental and biotechnological interest as M13 is a toolkit for numerous applications. I have one main issue and a few minor points to raise to the authors attention. English can be improved by a careful proofreading (some examples are listed below). Methods quality and description are fine.

Main issue:

1. Conformations of DNA in the viral particle. The authors claim in the Abstract (line 30), Results and Discussion that the single-stranded DNA is packed in the central region of the capsid in a conformation similar to an A-form dsDNA. This is somehow misleading as the A conformation is specific to dsDNA in which the antiparallel DNA chains are held together by Watson-Crick base pairs that are tilted relative to the helix axis in A-DNA. In case of the M13 structure the two non-pairing DNA strands are brought to build a helical arrangement due to interactions with pVIII but I see no evidence from the structural data that the two strands facing each other interact. In this sense, I would not name this case an A-form dsDNA conformation but simply a helical arrangement imposed by DNA strands-pVIII interaction whose topology parameters approach those of the dsDNA A-form. The authors could address/discuss the following questions:

- does the pVIII internal helical lattice display any periodic features (residues; charge...) that could explain how pVIII interacts/imposes the helical organization of the two independent DNA strands as it progressively covers DNA during assembly? A more detailed focus on the interaction pVIII-DNA appears very important to this referee as the driving mechanism for DNA folding.

- does the assembly process disrupt local DNA double-stranded bonding of the hairpins found in free circular M13 ssDNA that are displayed in Fig. S1A bottom schematics?

- is the resolution of the reconstruction enough to assume a perfect helical wrapping of the two DNA strands or is there latitude for local distortions as there is no base-pairing for stacking the bases of the two strands?

Details:

2. line 34 : ...between 1-2 μm

3. line 54: 18-amino acids (aa) signal peptide

4. line 55: 406-aa

5. Figure S1: describe abbreviation PS (packaging signal) in legend of panel A. It would help the reader to label the ssDNA sequences (e.g. in color) that form the hairpins displayed in the schematics underneath the sequence in A. Label lanes of gel and western blot in panels B and C, respectively. The scale bar in red of panel D is poorly visible.
6. Some of the particles in the cryo-micrograph of Fig S1D have some flexibility in the central helical rod (those particles were discarded during processing). Does the structure the pVIII filament provide a clue for its tilting/flexibility properties?
7. line 77: ...that provides...
8. lines 124-125; 133-138; 187-191; 203-204; 206...: the details of bonding between residues could be better compiled in an expanded supplementary Table 2 rendering the text more fluid.
9. line 153: layer 3
10. lines 168 to 172 and Fig. S5: very nice that the authors validated the structural data by a mutagenesis experiment
11. line 178: use either wild type phage (best) or normal phage
12. line 217: not sure the matryoshka doll analogy is suitable as it suggests several identical structures of decreasing size are packed ones inside the others
13. lines 236 to 242: not very clear
14. lines 244-245: Table S2 does not report on binding affinity. It illustrates only more extensive bonding.
15. line 257: not very clear where we shall look in Figure S2

Reviewer #1 (Remarks to the Author):

This is an impressive paper that provides new insights into the structure of filamentous bacteriophage which package ssDNA. A number of the observations are quite novel, such as the structure of the DNA in the phage. There are two major problems that need to be addressed in a major revision. One is that the paper needs to be carefully proofread by someone who speaks English as a native language. The other is that the discussion of the DNA structure could be expanded.

We appreciate the comments from the referee. We have sent the manuscript to a native speaker for careful proofreading. In addition, we have expanded the discussion by following the suggestions from the referees.

I began by indicating corrections needed, but gave up after the first page of the Introduction:

Line 38) “by thousands copies” should be “from thousands of copies”

Line 49) “function” should be “functioning”

Line 49) “which leads the insertion” should be “which directs the insertion”

Line 58) “have been extensively studies” should be “have been extensively studied”

Thanks! We have modified these sentences accordingly.

On Line 61, “due to the extreme shape of the virion” makes no sense. I assume the authors are referring to the fact that the minor proteins are only found at the ends and thus would be lost in conventional helical structure determination.

Thanks! We have removed this sentence.

Line 184, “could improve the overall resolution” should be “improved the overall resolution” , and the two have very different meanings.

Thanks! We have changed this sentence.

The parameters for A- and B-form DNA are treated with great precision, such as “A-form dsDNA, which has a rise of 28.6 Å per turn” and in Fig. 7 B-DNA is labeled as 35.7 Å. In fact, there is an enormous literature on the considerable variability of these parameters for both A- and B-form. Further, they really should say the pitch here, and not the rise, as the rise is best used for describing the base pairs. Surprisingly, there is no discussion of how dsDNA has been observed packed in the A-form in a number of filamentous archaeal viruses, such as in Wang et al., PNAS 117, 19643 (2020). In fact, to the best of my knowledge, there are no structures of B-form dsDNA in any filamentous virus. While the authors cite micro-mechanical stretching of B-DNA in vitro, there are also many observations of proteins stretching B-DNA, such as the bacterial RecA protein (for example, Yang et al., Nature 586, 801, 2020).

Thank you for your suggestions. We appreciate the suggestion and have replaced “rise” with the term "pitch" in the revision. For the parameters of the dsDNA, we changed these to ~28 Å and ~36 Å, respectively.

We fully agree with the referee’s point that there are no structures of B-form dsDNA reported in any filamentous virus. We acknowledge this point in the revision. We have added citations for the observations of proteins stretching B-DNA. In addition, we did a careful analysis and comparison, and had a detailed discussion on the possible form of the ssDNA at the bottom segment of our reconstruction. Given that the ssDNA structure could be

determined in our reconstruction, the capsid and the ssDNA should have a fixed relationship. As suggested by the referee 2, unlike the dsDNA, the double stranded helix structure of the ssDNA should be maintained mainly by the capsid rather than the paired bases. Four positively charged residues of the major capsid protein p8, including Lys40, Lys43, Lys44 and Lys48, are exposed in the lumen of the capsid and are supposed to have interactions with the phosphate-sugar backbone of the DNA. We placed and rotated an A-form dsDNA in the capsid with a step size of 1 degree and checked the possible interactions between the O3, OP1 and OP2 groups of the phosphate and the inner positively charged residues of the capsid at each rotation step. A plot of the contact numbers against the rotation angle clearly indicates that the ssDNA phosphate-sugar backbones in an A-form dsDNA arrangement could establish extensive interactions with the capsid (Figure 8). Similarly, the ssDNA could also establish extensive interactions with the capsid in the bottom segment, when adopting a conformation that has similar helical parameters as these of a B-form dsDNA. However, the ssDNA at the bottom could not adopt a conformation similar to that of a p form dsDNA as observed in the bacteriophage Pf1, in which the positively charged residues have extended conformation in the lumen and have direct interactions with the phosphate groups in the center (Figure S7). In our structure, the Lys residues in the lumen of the bottom segment maintain similar conformation as these in the middle part and none of these Lys residues have an extended conformation. The Lys residues in the current conformation could only establish possible interactions with the base groups rather than the phosphate groups of the p-form DNA in the center (Figure S7). While the phosphate-sugar backbones of the p-form DNA in the center are in an extremely extended state, it could not be stable without the direct contacts from the capsid. Furthermore, the nucleotide/pVIII ratio of Pf1 is 1.0, while the nucleotide/pVIII ratio of M13 and the mini variant of M13 is in the range of 2.1-2.4, indicating that the ssDNA of M13 is in a more compressed state. By

combining the data, we concluded that although so far no B-form dsDNA-like structures were observed in any filamentous virus, ssDNA in a double helix conformation that has similar parameters as a stretching B-DNA could be possible in a local region of some filamentous phages.

Reviewer #2 (Remarks to the Author):

This paper describes the cryoEM structure of a mini-variant of the filamentous phage M13. The structure includes all the minor proteins of M13 and of the major coat protein pVIII. The ingenious trick to use an M13 mini-virus with a short filament allowed solving the complete structure of the virus. The structure uncovers the molecular details how the filament 5-fold symmetric pVIII lattice is capped by different proteins at its two extremities. The work is beautiful and an important step forward to understand M13 structure and function. This is of significant fundamental and biotechnological interest as M13 is a toolkit for numerous applications. I have one main issue and a few minor points to raise to the authors attention. English can be improved by a careful proofreading (some examples are listed below). Methods quality and description are fine.

Main issue:

1. Conformations of DNA in the viral particle. The authors claim in the Abstract (line 30), Results and Discussion that the single-stranded DNA is packed in the central region of the capsid in a conformation similar to an A-form dsDNA. This is somehow misleading as the A conformation is specific to dsDNA in which the antiparallel DNA chains are held together by Watson-Crick base pairs that are tilted relative to the helix axis in A-DNA. In case of the M13 structure the two non-pairing DNA strands are brought to build a helical arrangement due to interactions with pVIII but I see no evidence from

the structural data that the two strands facing each other interact. In this sense, I would not name this case an A-form dsDNA conformation but simply a helical arrangement imposed by DNA strands-pVIII interaction whose topology parameters approach those of the dsDNA A-form. The authors could address/discuss the following questions:

- does the pVIII internal helical lattice display any periodic features (residues; charge···) that could explain how pVIII interacts/imposes the helical organization of the two independent DNA strands as it progressively covers DNA during assembly? A more detailed focus on the interaction pVIII-DNA appears very important to this referee as the driving mechanism for DNA folding.

Thank you for the comments. We fully agree with the referee's comments. We missed this part in the original version of the manuscript. Now we have analyzed the periodic features of the capsid and the interactions between the ssDNA genome and the capsid. We tried different strategies for detecting the interactions between the ssDNA genome and the capsid. By placing an A-form dsDNA model in the capsid, we were able to detect extensive interactions between the capsid and the DNA. The DNA could have a larger number of contacts at some positions when compared with these at other positions. We found that determination of the precise interactions between the ssDNA and the capsid depends on the accurate positions of the phosphate groups of the ssDNA and the conformation of the inner surface Lys residues, which, as the asymmetric reconstruction showing, have induced conformational changes upon the binding of ssDNA. However, due to the limited map resolution of the ssDNA part, we were not able to build a model for the ssDNA. The resolution limitation of the ssDNA part could be a result of flexibility in some of the nucleotides since the lack of base pair interactions. The conformation of the genome could be determined by complex factors, including the preformed conformation in the genome-pV complex, the

sequence of the genome, and the interactions with the coat proteins. We have added new sections in the results and discussions in the revision as the following:

New section in results:

“Interactions between the capsid and the ssDNA genome

The inner surface of the capsid is decorated with the positively charged residues K40, K43, K44 and K48, which are clustered in one region (Figure 8A). The distribution of the positively charge region in the capsid follows the symmetry of the capsid, which, however, is not consistent with that of either the A- or B-form dsDNA. Analysis of the asymmetric density map of the middle segment showed that K43, K44 and K48 of some pVIIs in layers 8-16 have significant asymmetric features and are in close proximity to the densities of the ssDNA (Figure 8B), suggesting possible direct interactions of these Lys residues with presumably the phosphate groups of the ssDNA phosphate-sugar backbone. The asymmetric conformation also indicates possible induced conformational changes of the inner surface Lys residues upon binding the ssDNA. To figure out possible interactions between the ssDNA genome and the capsid, we placed an A-form dsDNA structure of 4 turns in the capsid of the middle segment that contains layers 8-16 of pVIII. The length of the DNA (114.4 Å) is approximately the same as that of the capsid segment (114.8 Å). Then, the dsDNA model was rotated around the long axis with a step size of 1 degree. Close contacts ($< \text{or} = 4 \text{ Å}$) between the NZ atoms of the inner surface Lys residues and the OP1, OP2 and O3 atoms of the phosphate groups were counted at each rotation position (Figure 8C). The results showed that the average number of contacts at each rotation position is ~100. At some positions, the ssDNA has a larger number of contacts with the capsid. However, the number of contacts varies when a symmetric capsid structure built with the helical averaged map was used (Figure 8C). Similarly, a B-form-like dsDNA placed in the capsid can also establish extensive interactions with the capsid. Analysis of the 5-fold

symmetry-averaged density maps of the bottom and top segments, where the DNA adopts significantly different conformations compared to that in the middle segment, revealed that the Lys residues adopt similar conformations as those in the middle segment (Figure S7A). In the asymmetric reconstruction of the bottom segment, some of the Lys residues also display asymmetric features. These data combined suggest that the capsid is flexible for adapting genome DNA in various different conformations through probably induced conformational changes of the inner surface Lys residues. In our reconstruction, the precise positions of the phosphate groups of the ssDNA can not be determined due to the limited map resolution. The precise relative position of the ssDNA genome to the capsid could be further investigated through other techniques, such as solid NMR. “

In discussion:

“Unlike tailed bacteriophages, genome packaging of M13-like filamentous bacteriophages does not involve a preformed empty capsid. Instead, the ssDNA genome of M13 forms a complex with the non-structural protein pV in the cytoplasm. The packaging signal is exposed and not covered by pV, making it ready to be sensed by the top cap complex. However, the packaging signal in the top segment should have a different conformation from that observed in the absence of the top cap complex, as the stretching conformation of the ssDNA genome is unstable without this complex. Similarly, the A-form dsDNA-like or the stretching B-form dsDNA-like double-helix structure could not be stable without the capsid. In the process of genome packaging, the major coat protein pVIII replaces pV and interacts with the ssDNA in a channel embedded in the membrane. The conformation of the genome could be determined by complex factors, including the preformed conformation in the genome-pV complex, the sequence of the genome, and the interactions with the coat proteins.”

In addition, we added one new figure (Figure 8) to show the periodic feature of the inner surface of the capsid and the interactions between the ssDNA genome and the capsid.

- does the assembly process disrupt local DNA double-stranded bonding of the hairpins found in free circular M13 ssDNA that are displayed in Fig. S1A bottom schematics?

Thank you for the comment. After replication, the ssDNA is coated by pV proteins to form the pV-ssDNA complex, with the packing signal exposed and uncoated. The predicted secondary structure of the packing signal is a hairpin, while the tertiary structure of the packaging signal in the capsid likely has the extended phosphate backbone that can not be stable without the capsid. Thus, we think that during the assembly process, the pVII and pIX interact with the packing signal and may change the conformation of the packing signal. We have added this analysis in the discussion:

“Unlike tailed bacteriophages, genome packaging of M13-like filamentous bacteriophages does not involve a preformed empty capsid. Instead, the ssDNA genome of M13 forms a complex with the non-structural protein pV in the cytoplasm. The packaging signal is exposed and not covered by pV, making it ready to be sensed by the top cap complex. However, the packaging signal in the top segment should have a different conformation from that observed in the absence of the top cap complex, as the stretching conformation of the ssDNA genome is unstable without this complex. Similarly, the A-form dsDNA-like or the stretching B-form dsDNA-like double-helix structure could not be stable without the capsid. In the process of genome packaging, the major coat protein pVIII replaces pV and interacts with the ssDNA in a channel embedded in the membrane. The conformation of the genome could be determined by complex factors, including the

performed conformation in the genome-pV complex, the sequence of the genome, and the interactions with the coat proteins.”

-

Details:

2. line 34 : ...between 1-2 μm
3. line 54: 18-amino acids (aa) signal peptide
4. line 55: 406-aa

Thanks! We have corrected these.

5. Figure S1: describe abbreviation PS (packaging signal) in legend of panel A. It would help the reader to label the ssDNA sequences (e.g. in color) that form the hairpins displayed in the schematics underneath the sequence in A. Label lanes of gel and western blot in panels B and C, respectively. The scale bar in red of panel D is poorly visible.

Thank you for your suggestions. We have updated Figure S1 accordingly. In the legend of panel A, we will describe the abbreviation PS (packaging signal) and label the ssDNA sequences that form the hairpins displayed in the schematics underneath the sequence in A. We have also labeled the lanes of the gel and western blot in panels B and C, respectively. Additionally, we

adjust the scale bar in panel D to make it more visible.

6. Some of the particles in the cryo-micrograph of Fig S1D have some flexibility in the central helical rod (those particles were discarded during processing). Does the structure the pVIII filament provide a clue for its tilting/flexibility properties?

Created with BioRender.com.

Thank you for your suggestions. The cylindrical portion of the M13 capsid is flexible as indicated by the 2D classification, in which some classes of the M13 miniphage show small curvatures in the middle part of the particles. We tried to make a reconstruction. However, we failed to obtain a reasonable reconstruction with these particles, due to probably heterogenous conformations in these particles. As shown in the pVIII of layers 3-6, pVIII can

bend with different curvatures. We do believe the flexibility of the pVIII and the fish-scale like arrangement of pVIII would allow bending in the capsid.

7. line 77: ...that provides...

Thanks! Corrected.

8. lines 124-125; 133-138; 187-191; 203-204; 206...: the details of bonding between residues could be better compiled in an expanded supplementary Table 2 rendering the text more fluid.

Thanks! We have put these detailed information in the supplementary Table S2.

9. line 153: layer 3

Thanks! Corrected.

10. lines 168 to 172 and Fig. S5: very nice that the authors validated the structural data by a mutagenesis experiment

Thank you for your comments.

11. line 178: use either wild type phage (best) or normal phage

Thanks! Corrected.

12. line 217: not sure the matryoshka doll analogy is suitable as it suggests several identical structures of decreasing size are packed ones inside the others

Thanks! We have removed the Matryoshka doll and have revised the description of this part.

13. lines 236 to 242: not very clear

Thanks! We have revised this part, moved Figure 1D to Figure 7B, and have made a new figure panel (Figure 7C) to make this part clear.

14. lines 244-245: Table S2 does not report on binding affinity. It illustrates only more extensive bonding.

Thank you for indicating this. We have revised this part.

15. line 257: not very clear where we shall look in Figure S2

Thanks! We have modified this.

In addition to all these changes mentioned above, we have moved Figure 1C to Figure 2 to make the description more clear.

REVIEWERS' COMMENTS

Reviewer #2 (Remarks to the Author):

I find the manuscript was significantly improved. In particular, the more detailed analysis of the circular ssDNA conformations and its interactions with capsid proteins in the viral particle adds to the scientific interest of this exciting work. Although the resolution does not allow yet mapping individual nucleotide-amino acid interactions, the originality of the protein-ssDNA interactions is of interest and will likely inspire further work in this system.

I still have some comments and a number of minor points to be addressed. English would still benefit from improvement.

Major point:

1. (lines 315-328) In the new part of the manuscript on capsid-ssDNA interactions the authors modelled a A-form dsDNA in the middle segment of the viral particle and report on close capsid-DNA contact upon rotation of the DNA helix in the asymmetric (Fig. 8C, top) and symmetric (Fig. 8C, bottom) capsid reconstruction. I can imagine what the authors have in mind but it is not clear from the text (i) what is the rationale of this comparison and (ii) what does it tell about the protein-DNA interaction. Please clarify these points.

To "aerate" the text for the reader I would suggest making a full stop and starting a new paragraph to describe the analysis on the interactions in the top and bottom segments of the viral particle. Finally, I would clearly state that symmetrized structures are not suitable to unravel the protein-DNA interactions between the symmetry mismatched capsid and DNA.

Minor points:

2. Figure S1. Panel A: change "packing signal" to "packaging signal" in panel A for coherence with the text. Panel B: describe in the figure legend the abbreviations S, W, RaW, E, RaE, CE used on top of the gel lanes; there is no red arrow in panel B (add it or remove statement from figure legend); why do you use a dashed red rectangle in the figure?

3. line 134-135: rephrase "Approximately 1/3 of the pIX helix C-terminal is straight and aligns almost..."

4. lines 135-136: indicate angle of pIX helix bending and specify which conformer of pVIII has an helix with similar curvature (pVIII in the particle middle or in the caps?)

5. line 167: "consists of 20 protein layers..."

6. lines 192 and 196: Figure 4A shows neither pVIII-pIII interactions nor pIII-pVI interactions. Those are shown in Fig. 1B and, best, in Fig.5 for pIII-pVI. Change figure citations.
7. line 201: could cite Fig. 5A.
8. line 203: "...through hydrophobic residues..."
9. lines 204-205: information within brackets can be eliminated as those details are listed in table S2.
10. Fig. S5 legend. I would specify in the legend "Phage titers produced by M13KO7 mutant genomes" and orient the reader to Materials and Methods for the experimental setup to understand how the lacZ activity scores for viable phages.
11. line 305: "residues K40, K43, K44 and K48, which are clustered in one region of pVIII"
12. line 339: "solid state NMR"
13. line 370: "with the non-structural protein pV"
14. line 535: Figure S5 (not S4).
15. line 774: correct: "the two positively charged rings of pVIs."
16. Fig. 6C and legend: label the orange helix on the left panel and the yellow loop (L24 pVI') in the central panel. Correct the legend (lines 791-794): left and right is not clear. There are three panels. Name them left, centre (or middle), right and describe the content of each of them.
17. lines 802-803: "A schematic diagram of the DNA structure model is shown on the right"
18. legend of Fig. S6: it is more logical to describe layers from layer 9 to layer 15
19. Fig. 7. It is important to specify if this is the symmetrized particle of the asymmetric particle structure. From visual inspection of the lysine residues, it looks to be C5 symmetrized. If this is the case is there a reason to not show the asymmetric structure?
20. It could be justified to cite the work of Liu and Day (1994) Science 265: 671-674 (<https://doi.org/10.1126/science.8036516>) on the Pf1 virus structure

I find the manuscript was significantly improved. In particular, the more detailed analysis of the circular ssDNA conformations and its interactions with capsid proteins in the viral particle adds to the scientific interest of this exciting work. Although the resolution does not allow yet mapping individual nucleotide-amino acid interactions, the originality of the protein-ssDNA interactions is of interest and will likely inspire further work in this system.

We appreciate the comments and suggestions from this referee. These are very helpful.

I still have some comments and a number of minor points to be addressed. English would still benefit from improvement.

Major point:

1. (lines 315-328) In the new part of the manuscript on capsid-ssDNA interactions the authors modelled a A-form dsDNA in the middle segment of the viral particle and report on close capsid-DNA contact upon rotation of the DNA helix in the asymmetric (Fig. 8C, top) and symmetric (Fig. 8C, bottom) capsid reconstruction. I can imagine what the authors have in mind but it is not clear from the text (i) what is the rationale of this comparison and (ii) what does it tell about the protein-DNA interaction. Please clarify these points.

To “aerate” the text for the reader I would suggest making a full stop and starting a new paragraph to describe the analysis on the interactions in the top and bottom segments of the viral particle. Finally, I would clearly state that symmetrized structures are not suitable to unravel the protein-DNA interactions between the symmetry mismatched capsid and DNA.

Thank you for the comments! We have made it clear in the text by adding the following sentences after the description of the interactions between the DNA and the symmetric capsid: **“Furthermore, the plot calculated with the symmetric capsid shows no distinct peaks, indicating that the capsid with a uniform and symmetric conformation is unlikely able to establish a unique interaction with the ssDNA and trap the ssDNA at a specific position.”**

We have revised the manuscript accordingly and started a new paragraph to describe the analysis of the interactions in the top and bottom segments.

Minor points:

2. Figure S1. Panel A: change “packing signal” to “packaging signal” in panel A for coherence with the text. Panel B: describe in the figure legend the abbreviations S, W, RaW, E, RaE, CE used on top of the gel lanes; there is no red arrow in panel B (add it or remove statement from figure legend); why do you use a dashed red rectangle in the figure?

Thank you for the comments. We have corrected the “packing signal” to “packaging signal” in Figure S1 for coherence with the full text.

We have added the arrow pointing the pVIII band on the gel. In addition, we have put the correct version of this panel showing the SDS-PAGE gel analysis of the purified miniphage. In the previous version, we showed the uncropped gel and the red dash box was used for indicating the final result that should be presented in the supplementary figure. We have put the correct form of the panel in the figure and have moved the rawdata into the source data file.

3. line 134-135: rephrase “Approximately 1/3 of the pIX helix C-terminal is straight and aligns almost…”

4. lines 135-136: indicate angle of pIX helix bending and specify which conformer of pVIII has an helix with similar curvature (pVIII in the particle middle or in the caps?)

Thanks! We have rephrased this sentence and have indicated angle of pIX helix bending. The revised sentence is “The C-terminal region of the pIX helix, comprising approximately one-third of the helix, is straight and nearly parallel to the main axis of the virion. The helix of pIX bends around residue W15, introducing a curvature similar to those of the pVIII helices¹⁵ (Figure 1A). The bend creates an angle of approximately 13° between the N-terminal portion of the pIX helix and the main axis of the virion.”

5. line 167: “consists of 20 protein layers…”

Thanks! We have corrected this.

6. lines 192 and 196: Figure 4A shows neither pVIII-pIII interactions nor pIII-pVI interactions. Those are shown in Fig. 1B and, best, in Fig.5 for pIII-pVI. Change figure citations.

Thanks! We have changed the figure citations here.

7. line 201: could cite Fig. 5A.

Thanks! We have added this figure citation.

8. line 203: “…through hydrophobic residues…”

Thanks! We have corrected this.

9. lines 204–205: information within brackets can be eliminated as those details are listed in table S2.

Thanks! We have deleted the information within the bracket.

10. Fig. S5 legend. I would specify in the legend “Phage titers produced by M13KO7 mutant genomes” and orient the reader to Materials and Methods for the experimental setup to understand how the lacZ activity scores for viable phages.

Thanks! We have changed the legend of figure S5 and added “More details could be found in Methods: Plaque assay of the M13KO7 mutants” to orient the reader.

11. line 305: “residues K40, K43, K44 and K48, which are clustered in one region of pVIII”

Thanks! We have corrected this and added “of pVIII” in the revised version.

12. line 339: “solid state NMR”

Thanks! We have corrected this.

13. line 370: “with the non-structural protein pV”

Thanks! We have corrected this.

14. line 535: Figure S5 (not S4).

Thanks! We have corrected this.

15. line 774: correct: “the two positively charged rings of pVIs.”

Thanks! We have corrected this.

16. Fig. 6C and legend: label the orange helix on the left panel and the yellow loop (L24 pVI') in the central panel. Correct the legend (lines 791–794): left and right is not clear. There are three panels. Name them left, centre (or middle), right and describe the content of each of them.

Thanks! We have labeled the orange helix (Middle Ln+2 pVIII) and yellow loop (Layer 25 pIII) in the left and central panel of Figure 6C. We also corrected the legend of Figure 6C as "(C) Left and Middle: Ribbon diagrams and structural comparisons of the pockets with pVIII (left) and with pIII (middle). Right: Ribbon diagrams showing the comparisons of the key residues in the pocket that accommodates pIII with these in the pocket that accommodates pVIII."

17. lines 802-803: "A schematic diagram of the DNA structure model is shown on the right"

Thanks! We have modified this part according to the suggestion.

18. legend of Fig. S6: it is more logical to describe layers from layer 9 to layer 15

Thanks! We changed "Interactions in the helical assembly of the M13 pVIII capsid" to "Interactions among pVIIIs in layer 9 to layer 15 of the capsid."

19. Fig. 7. It is important to specify if this is the symmetrized particle of the asymmetric particle structure. From visual inspection of the lysine residues, it looks to be C5 symmetrized. If this is the case is there a reason to not show the asymmetric structure?

Thanks! We thought you mean the Figure S7. This is a longitudinal comparison of different pVIIIs. In this figure, we mainly want to illustrate that the Lys residues in different segment have similar conformations and can adapt different conformations of the ssDNA, including the A/B or stretched B dsDNA-like conformations. However, the conformation of the Lys residues should not be suitable for adapting the P-form-like DNA as that require a different conformation of the Lys that points towards the center of the particle. Through these comparisons and analysis, we want to make a conclusion that the P-form-like DNA structure should not exist in the M13 miniphage. We have added in the figure caption "The comparisons indicate that the overall orientations of the Lys residues in different segments are similar. It would require the change of the orientation for the Lys residues to establish interactions with some DNA forms, such as a P-form DNA. The symmetric capsid structures were used for the figure preparations."

20. It could be justified to cite the work of Liu and Day (1994) Science 265: 671-674 (<https://doi.org/10.1126/science.8036516>) on the Pf1 virus structure

Thanks! We have cited this paper in text at line 440.